# Dynamic Phenomena and Complexation Effects in the α-Lithiation and Asymmetric Functionalization of Azetidines

**DOI:** 10.3390/molecules27092847

**Published:** 2022-04-29

**Authors:** Pantaleo Musci, Marco Colella, Angela Altomare, Giuseppe Romanazzi, Nadeem S. Sheikh, Leonardo Degennaro, Renzo Luisi

**Affiliations:** 1Department of Pharmacy—Drug Sciences, University of Bari “A. Moro”, Via E. Orabona 4, 70125 Bari, Italy; pantaleo.musci@uniba.it (P.M.); marco.colella@uniba.it (M.C.); 2National Research Council (CNR), Institute of Christallography IC-CNR, Via Amendola 127/A, 70125 Bari, Italy; angela.altomare@ic.cnr.it; 3DICATECh—Dipartimento di Ingegneria Civile, Ambientale, del Territorio, Edile e di Chimica, Politecnico di Bari, Via Orabona 4, 70125 Bari, Italy; giuseppe.romanazzi@poliba.it; 4Chemical Sciences, Faculty of Science, Universiti Brunei Darussalam, Jalan Tungku Link, Gadong BE1410, Brunei Darussalam

**Keywords:** azetidines, organolithiums, nitrogen dynamics, heterocyclic chemistry, computational chemistry

## Abstract

In this work it is demonstrated that enantiomerically enriched *N*-alkyl 2-oxazolinylazetidines undergo exclusive α-lithiation, and that the resulting lithiated intermediate is chemically stable but configurationally labile under the given experimental conditions that afford enantioenriched *N*-alkyl-2,2-disubstituted azetidines. Although this study reveals the configurational instability of the diastereomeric lithiated azetidines, it points out an interesting stereoconvergence of such lithiated intermediates towards the thermodynamically stable species, making the overall process highly stereoselective (er > 95:5, dr > 85:15) after trapping with electrophiles. This peculiar behavior has been rationalized by considering the dynamics at the azetidine nitrogen atom, the inversion at the C-Li center supported by in situ FT-IR experiments, and DFT calculations that suggested the presence of η^3^-coordinated species for diastereomeric lithiated azetidines. The described situation contrasted with the demonstrated stability of the smaller lithiated aziridine analogue. The capability of oxazolinylazetidines to undergo different reaction patterns with organolithium bases supports the model termed “dynamic control of reactivity” of relevance in organolithium chemistry. It has been demonstrated that only 2,2-substituted oxazolinylazetidines with suitable stereochemical requirements could undergo C=N addition of organolithiums in non-coordinating solvents, leading to useful precursors of chiral (er > 95:5) ketoazetidines.

## 1. Introduction

Azetidines [1] are particularly interesting structural motifs among the large family of saturated nitrogen heterocycles. Although azetidines have been regarded as esoteric analogues of aziridines, recently such four-membered heterocycles have proven to be appealing for diverse applications, especially in medicinal chemistry and as agrochemicals. The interest towards this heterocyclic scaffold culminated in the development of new strategies for its synthesis. These include ring contractions, cycloadditions, C–H activations, cross-couplings, and strain-release tactics [2,3,4,5,6,7]. Among the different strategies to synthesize azetidines, the base promoted C2 metalation with subsequent electrophilic trapping is a valid approach for accessing the more complex azetidine derivatives that start from a preformed heterocyclic core. In this context, recent studies on *N*-protected azetidines have shed some light on the structural factors that play a key role in the regio- and stereoselective C2-lithiation reaction. In fact, when an electron-withdrawing group, such as *t*-BuOC=O (Boc), *t*-BuSO_2_ (Bus), *t*-BuOC=S (thio-Boc), or thiopivaloyl (*t*-BuC=O), is installed on the azetidine’s nitrogen, exclusive α-lithiation occurs and a variety of C2-functionalized azetidines can be prepared via the α-lithiation/electrophilic trapping sequence (Figure 1a) [8,9,10,11,12]. Additional insights have been provided by our group on the factors affecting the reactivity and regioselectivity of the lithiation of *N*-substituted aziridines and azetidines [13,14,15]. Strained aza-heterocycles nitrogen dynamics and complexation phenomena play a pivotal role in controlling the regio- and stereoselectivity of the lithiation reaction [16]. Interestingly, while *N*-Boc-2-aryl-azetidines underwent exclusive α-lithiation upon reaction with a lithium base, [17,18] the presence of an electron-donating group (EDG), such as an alkyl group, on the nitrogen atom renders the azetidine ring—an effective ortho-directing metalation group (DMG)—able to promote exclusive aromatic lithiation (Figure 1a) [19]. In fact, an *N*-EDG group would increase either the basicity or the coordinating capability of the nitrogen. Therefore, a drastically enhanced kinetic acidity of the ortho-aromatic protons is observed jointly with a higher capability for the nitrogen atom to act as an ortho-directing group. These results prompted us to propose a model based on the assumption that dynamic factors and coordination effects could play an important role in addressing the regioselectivity of the lithiation of small heterocycles. It could be claimed that the control of molecular stereodynamics by external stimuli could be exploited to prepare structurally different compounds from the same starting material because of a different reactivity related to a configuration or conformational preference. This intriguing concept that we termed “dynamic control of reactivity” has already been demonstrated in our group some years ago while studying the C-H functionalization of aziridines bearing an *N*-EDG substituent that was able to undergo rapid *N*-inversion (Figure 1b). A control on the rate of the nitrogen inversion by controlling the reaction temperature allowed for switching between two different reaction pathways (i.e., ortho-lithiation or α-lithiation) [20,21,22]. Similarly, the complexation capability of alkylideneaziridines was affected by the solvent, thereby allowing for a stereochemical switch in the lithiation reaction (Figure 1b) [23].

More recently, the study on aziridines bearing an EDG group (i.e., alkyl) on the nitrogen atom, and an oxazoline moiety as the EWG group at C2 revealed the importance of complexation and nitrogen dynamics in their reactivity with organolithium reagents [24]. In particular, starting from optically active *N*-phenylethyl-2-oxazolinylaziridines, a regioselective lithiation was observed with *n*-butyllithium at −78 °C in coordinating solvent, such as THF, thereby furnishing enantioenriched 2,2-disubstituted aziridines (Figure 2a). The same study demonstrated that the configurational stability of the α-lithiated intermediate was due to a high barrier to *N*-inversion. Remarkably, the same aziridines underwent competitive reaction pathways in non-coordinating solvents, such as toluene. In competition with the deprotonation, a counterintuitive nucleophilic attack of the organolithium to the C=N double bond of the oxazoline ring was observed. Such a nucleophilic addition was the result of a combination of strict stereochemical requirements associated with the nitrogen inversion and complexation phenomena (Figure 2a). For the sake of comparison, a similar study was executed on the homologue *N*-alkyl-2-oxazolinylazetidine (Figure 2b) [25]. In striking contrast to the three membered heterocycles, this comparative study on *N*-alkyl-2-oxazolinylazetidine, which was conducted in a non-polar solvent such as toluene, revealed a more interesting reactivity profile. In fact, the exclusive chemoselective nucleophilic attack of the organolithium to the C=N bond of the oxazoline ring was observed, even at 0 °C, exclusively affording the corresponding oxazolidinylazetidine in a highly stereoselective manner. Remarkably, this approach led to hardly accessible chiral 2-acylazetidines in good yields and high enantiomeric ratios upon the hydrolysis of the oxazolidine ring under mild acidic conditions. The proposed mechanism for this stereoselective addition, which is supported by DFT calculations, relies again on strict stereochemical requirements and complexation phenomena.

With these results in hand, we felt compelled to further explore the reactivity of *N*-alkyl oxazolinylazetidines in more polar solvents to check if the α-lithiation could have been a possible reactive event (Figure 2c). We report herein the result of this investigation and a comprehensive spectroscopic and computational study on the lithiated species involved in this highly stereoselective process.

## 2. Results

Diastereomeric oxazolinyl azetidines (2*S*,1′*R*)-**1** and (2*R*,1′*R*)-**1** were prepared starting from chiral diastereomeric *N*-[(*R*)-1-phenylethyl)azetidine-2-carboxylic acid esters (see Appendix A). First, we investigated the stereoselectivity for a deuteration reaction by reacting chiral (2*R*,1′*R*)-**1** with organolithium bases in THF at low temperatures, followed by electrophilic quench (Table 1). As expected, when a solution of (2*R*,1′*R*)-**1** was reacted with 1.2 equivalents of *n*-hexyllithium (*n*-HexLi) at −78 °C for 30 min, followed by quenching with a deuterium source, a mixture of diastereomeric deuterated products (2*R*,1′*R*)-**2a** and (2*S*,1′*R*)-**2a** were obtained, albeit with a low conversion (Table 1, entry 1). To our delight, better yields and higher conversions were observed by using 2.2 and 2.8 equivalents of *n*-HexLi (entries 2 and 3), albeit reaction conversion was up to 40%. This prompted us to further optimize the reaction conditions and, gratifyingly, when the reaction was carried out with 3.5 equivalents of *n*-HexLi, (2*R*,1′*R*)-**2a** was obtained in a 90% yield along with 10% of (2*S*,1′*R*)-**2a** (entry 4).

Under the optimized conditions (2 equivalents of *s*-BuLi, 20 min, −78 °C), the ^1^H NMR analysis of the crude reaction mixture showed the formation of diastereomeric deuterated products, (2*R*,1′*R*)-**2a** and (2*S*,1′*R*)-**2****a**, in a 90:10 ratio (Figure 1). This result suggested that azetidine (2*R*,1′*R*)-**1** could epimerize under the lithiation conditions, which are likely for the intrinsic configurational lability of the lithiated intermediate [26]. Similarly, the lithiation/deuteration sequence on diastereomeric (2*S*,1′*R*)-**1**, under optimized conditions (Table 1, entry 6), furnished a diastereomeric mixture of deuterated azetidines (2*R*,1′*R*)-**2a** and (2*S*,1′*R*)-**2a** in a high yield and with the same diastereomeric ratio (90:10) observed for (2*R*,1′*R*)-**1** (Table 1). With this evidence in hand, it is reasonable to assume that both azetidines, (2*R*,1′*R*)-**1** and (2*S*,1′*R*)-**1,** could be regioselectively lithiated at the α-position (i.e., C2), and that the putative intermediates, (2*R*,1′*R*)-**1**-**Li** and (2*S*,1′*R*)-**1**-**Li,** could undergo fast equilibration (epimerization). In striking contrast to configurationally stable lithiated oxazolinylaziridines, lithiated oxazolinylazetidines undergo rapid epimerization that is likely by a double inversion of the configuration at the lithiated carbon and at the azetidine’s nitrogen.

In Figure 3, it is reported that the epimerization process involved the lithiated azetidines (2*R*,1′*R*)-**1-Li** and (2*S*,1′*R*)-**1-Li**. In particular, the process must involve four different species resulting, respectively, from two C-Li inversions and two N-R inversions. In fact, NOESY experiments confirmed that, in starting azetidines (2*R*,1′*R*)-**1** and (2*S*,1′*R*)-**1**, the oxazolinyl ring and the *N*-phenylethyl group are set in an anti-arrangement [25]. This suggests that starting from (2*R*,1′*R*)-**1**, upon deprotonation, *trans*-(2*R*,1′*R*)-**1-Li** could be generated first and then undergo either a C-Li inversion producing *syn*-(2*S*,1′*R*)-**1-Li** or N-R inversion giving *syn*-(2*R*,1′*R*)-**1-Li**. On this basis, N-R inversion on *syn*-(2*S*,1′*R*)-**1-Li** would give *anti*-(2*S*,1′*R*)-**1-Li**, which, upon C-Li inversion, would converge again to *syn*-(2*R*,1′*R*)-**1-Li** in a sort of looping process. This dynamic scheme would explain the 90:10 diastereomeric ratio and the stereo-convergence towards (2*R*,1′*R*)-**2a** that was observed in the lithiation/deuteration sequence of both (2*S*,1′*R*)-**1** and (2*R*,1′*R*)-**1**. However, the configurational lability observed in these lithiated azetidines is in striking contrast to the configurational stability observed for the corresponding oxazolinylaziridines (Figure 3b).

In fact, DFT calculations and reactivity studies on lithiated oxazolinylaziridines supported the hypothesis that the energy barrier for both N-R and C-Li inversion is too high to be overcome, thus contributing to the configurational stability of η^3^-coordinated lithiated intermediates [24]. By contrast, N-R inversion and ring puckering in azetidine is expected to be easier with respect to aziridines due to a lower energy barrier for the larger heterocycle [27,28]. With the aim to get useful insights for this epimerization process, we decided to perform an in situ FT-IR investigation and DFT calculations on the neutral and lithiated azetidines species involved in this process.

## 3. In Situ FT-IR Investigation

By using a mid-infrared FT-IR probe, the progression of the lithiation reaction was monitored in situ at −78 °C [29,30,31]. The investigation was restricted to the range of 1700–1575 cm^−1^, which includes the diagnostic stretching vibrations of the C=N double bond of the oxazoline ring. In neutral azetidine (2*R*,1′*R*)-**1**, the C=N signal was detected at 1658 cm^−1^ in 0.2 M THF solution at −78 °C (Figure 2). In the first experiment, upon addition of a solution of *s*-BuLi, new signals were observed in the range of 1580–1625 cm^−1^. The new signals were ascribed to the putative lithiated azetidines (Figure 2a). Monitoring the progress of the reaction for 20 min revealed the complete disappearance of the signal at 1658 cm^−1^ and the persistence of signals in the range of 1580–1625 cm^−1^. No further changes were observed when monitoring the reaction for 1 h. The newly detected signals were centered at 1603 cm^−1^ and ascribed to the equilibrating lithiated species (2*R*,1′*R*)-**1-Li** and (2*S*,1′*R*)-**1-Li**.

In a second in-situ FT-IR experiment using a faster signal detection, a transient large signal in the range of 1650–1590 cm^−1^ (Figure 2b) was observed a few seconds after the addition of the lithium base. According to previous evidence for similar reactions [32], we supposed that the transient signal could belong to a pre-lithiation complex between the oxazolinylazetidine and the organolithium base just before the deprotonative event. Moreover, deprotonation was concluded in 15–20 s, resulting in the disappearance of the signal at 1658 cm^−1^ and the appearance of a broad signal in the range of 1580–1625 cm^−1^ that was assigned to the equilibrating lithiated species (2*R*,1′*R*)-**1-Li** and (2*S*,1′*R*)-**1-Li**. When the electrophile (CD_3_OD) was added dropwise to the reaction mixture, the broad signals disappeared and a new signal at 1650 cm^−1^ (νC=N), which was likely the quenched oxazolinylazetidine, appeared. ^1^H NMR analysis of the crude reaction mixture confirmed the presence of azetidines (2*R*,1′*R*)-**2a** and (2*S*,1′*R*)-**2a** as a 90:10 mixture of diastereoisomers. It is worth mentioning that lithiation on diastereoisomer (2*S*,1′*R*)-**1**, conducted under in-situ FT-IR monitoring, returned the same IR profile observed for (2*R*,1′*R*)-**1**. This result supports the hypothesis that upon addition of *s*-BuLi, a configurationally labile lithiated intermediate is generated and a stereo-convergence favoring (2*R*,1′*R*)-**1**-**Li** takes place.

## 4. DFT Studies

To further support the experimental evidence obtained by the in-situ FT-IR analysis, as well as to get additional information on the structure and energies of the lithiated oxazolinylazetidines (2*R*,1′*R*)-**1-Li** and (2*S*,1′*R*)-**1-Li**, a detailed DFT study was executed. Neutral and lithiated oxazolinylazetidines were subjected to DFT computational analysis using different DFT methods and solvation models (see Supplementary Information). The relative stereochemistry (*syn* or *anti*) between the *N*-substituent and the oxazoline ring was also judiciously taken into consideration. For neutral azetidines (2*R*,1′*R*)-**1** and (2*S*,1′*R*)-**1**, NOESY experiments confirmed that the oxazoline ring and the *N*-substituent have an *anti*-arrangement. Pleasingly, the DFT calculations also corroborated this experimental stereochemical evidence, and a computed unscaled wavenumber for νC=N was in good agreement with the corresponding experimental values (Figure 3).

As delineated in Figure 3, four lithiated oxazolinylazetidines, likely involved in the epimerization process, were computationally investigated as well. The computed structures of *anti*-(2*R*,1′*R*)-**1-Li** at the WB97XD 6-31+G(d,p) level (Figure 4 and Appendix A) showed an important stabilization effect in the solvent THF. Moreover, both the IEF-PCM and CPCM solvation models revealed an interesting relationship for the computed wavenumber for the C=N bond and the experimental values and furnished the Gibbs free energies for the four lithiated intermediates. Lithium can be bonded either by the oxazoline ring (via η^3^-complex) or by the lone pair of the azetidine’s nitrogen for the anti and syn arrangement of the substituents, respectively (see infra). However, the DFT study disclosed that the most stable structure was *anti*-(2*R*,1′*R*)-**1-Li**, which showed an η^3^-complex and a calculated νC=N of 1613 cm^−1^ (using WB97XD 6-31+G(d,p) basis set and IEF-PCM model). This calculated C=N vibrational value appears consistent with the value of 1603 cm^−1^ that was observed by in situ FT-IR experiments, and at lower wavenumber with respect to the neutral oxazolinylazetidine *anti*-(2*R*,1′*R*)-**1** (νC=N_calc_ 1655 cm^−1^ vs. νC=N_exp_ 1658 cm^−1^). Interestingly, in agreement with the experimental observations, the more stable lithiated structures were *anti*-(2*R*,1′*R*)-**1-Li** and *anti*-(2*S*,1′*R*)-**1-Li**, which showed the involvement of the C=N group in the complexation of the lithium ion. This justifies the shift of ~50 cm^−1^ to lower wavenumbers for the C=N stretching frequency upon lithiation, and it is indicative of a weakening of the C=N bond that is likely involved in the η^3^-coordination with lithium (Figure 4).

## 5. Reaction Scope

After addressing the structural features of the lithiated azetidines and assessing that a major stereoisomer could be involved, the synthetic potentials were explored. With the optimized conditions in hand, both diasteromeric oxazolinylazetidines (2*R*,1′*R*)-**1** and (2*S*,1′*R*)-**1**, were employed in the lithiation/electrophile trapping sequence, leading to products **2b-g** (Figure 4). The 2,2-disubstituted azetidines **2b-g** were isolated in moderate to excellent yields and with high stereoselectivity. In almost all cases a major distereoisomer was observed whose absolute stereochemistry at C2 was supposed to be (*R*), based on the result observed in the lithiation/deuteration experiments [33]. As reported in Figure 4, quenching with alkyl halides returned highly enantioenriched 2,2-disubstituted azetidine **2b-g** in good to excellent yields and high diastereoselectvity (dr > 85:15). By using Boc_2_O as the electrophile, enantioenriched ester **2h** was obtained in a 94% yield and 90:10 diastereomeric ratio. The use of acetone as a representative carbonyl electrophile resulted in lower diastereoselectivity (dr 65:35) for the trapping product **2i**. However, the two diastereoisomers were separable and were found highly enantioenriched (er > 99:1). The identity and stereochemistry of the diastereisomer ***minor***-**2i**, was ascertained by single crystal X-Ray diffraction [34]. The stereochemistry at C2 in ***minor***-**2i** agreed with the stereochemistry observed for the alkylation reactions. This would suggest that, although the α-lithiation produces equilibrating lithiated azetidines, the nature of the electrophile could affect the diastereoselectivity of the trapping reaction. The possibility of an electrophile-dependent kinetic resolution cannot be ruled out at this stage. Moreover, this process is complicated by the involvement of four different lithiated azetidines as reported in Figure 3. Interestingly, the alkylation reaction produced different *N*-stereoisomers depending on the steric demand of the alkyl substituent introduced at C2. In detail, 2D-NOESY experiments (see Appendix A) on methylated product **2b** clearly showed that the *N*-substituent and the oxazoline ring adopt an anti-arrangement. In striking contrast, 2D-NOESY experiments demonstrated that in derivatives **2c-g** the *N*-substituent and the oxazoline ring are in a syn-arrangement. The stereochemical preference has also been confirmed by DFT calculations for **2b** and **2c** (see Appendix A). Similarly, 2D-NOESY experiments for both diastereoisomers ***major*-2i** and ***minor*-2i** showed the *syn*-arrangement between the *N*-substituent and the oxazoline ring, as confirmed by the X-ray analysis of ***minor*-2i** (Figure 4). This stereochemical preference is relevant if we consider the epimerization mechanism reported in Figure 3. In fact, *syn*-(2*R*,1′*R*)-**1-Li** could be involved in the direct retentive trapping of the electrophile. However, at this stage, the possibility of *N*-inversion after the reaction with the electrophile of (2*R*,1′*R*)-**1-Li** cannot be ruled out.

The stereochemical preference (syn or anti) of the azetidine’s nitrogen substituent is also relevant for reactivity. In fact, the nucleophilic addition at the C=N bond of the oxazoline ring is strongly dependent on the stereochemistry of the oxazolinylazetidine [24]. According to our previous findings, which showcase that the nucleophilic addition of organolithiums to the C=N bond of the oxazoline ring occurs smoothly in toluene, we wondered if the same process could have been productive with 2,2-disubstituted oxazolinylazetidines en route to not easily obtainable 2,2-ketoazetidines. To our delight, the nucleophilic addition of ethyllithium and butyllithium to oxazolinylazetidine *anti*-(2*R*,1′*R*)-**2b** in toluene at −78 °C produced, after acidic work-up and silica gel chromatography, enantioenriched 2-acylazetidines (2*R*,1′*R*)-**3a,b** in very good yields and absolute preservation (er > 95:5, Figure 5a). In striking contrast, the same nucleophilic addition of organolithiums did not take place with oxazolinylazetidine *syn*-(2*R*,1′*R*)-**2c** bearing the oxazoline ring and the *N*-substituent from the same side (Figure 5b). In this last case, unreactive starting material was recovered. The key in these chemoselective additions to the C=N bond is in the right stereochemistry (i.e., *syn* or *anti*), which is able to produce reactive or unreactive complexes with the organolithium nucleophile (Figure 5). In further detail, only with the *anti*-arrangement—as for *anti*-(2*R*,1′*R*)-**2b**, for example—is it possible to produce a reactive complex that is able to promote C=N addition. The lack of reactivity of *syn*-(2*R*,1′*R*)-**2c** supports the model, based on complexation previously reported [25].

## 6. Conclusions

In conclusion, a method for accessing enantioenriched *N*-alkyl-2,2-disubstituted azetidines by α-lithiation of *N*-EDG azetidines has been reported. The study revealed that a chemically stable but configurationally labile lithiated azetidine is involved, which is in contrast with the reported configurational stability of the corresponding lithiated aziridines. The involvement of equilibrating diastereoisomeric lithiated azetidines has been supported by in-situ FT-IR analysis and DFT calculations. The importance of stereochemical preferences at the azetidine’s nitrogen has also been demonstrated with the enantioselective preparation of 2-acylazetidines.

## Data Availability

The data presented in this study are available in Appendix A.

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
