# Peer review of "Dynamic Phenomena and Complexation Effects in the α-Lithiation and Asymmetric Functionalization of Azetidines"

_molecules, 2022, doi:10.3390/molecules27092847_

Round 1

Reviewer 1 Report

  1. The quality of Scheme 1 is not good and must be improved.
  2. Is it possible to sudy the the progression of the lithiation reaction by dynamic NMR method?
  3. In theoretical calculations, it is unclear why the level of theory was chosen.  A reason for choosing this method should be presented here. 
  4. What is the scaling factor for calculated frequency for C=N bond?
  5. In-situ FT-IR investigation, what is the temperature?
  6. What is the effect of temperature on epimerization and inversion process? All investigations were done at -78 ˚C.
  7. The authors should be calculated chemical shifts of epimeric hydrogen in all stuctures and compare with experimental data, like computed wavenumber for C=N bond.

Author Response

1. The quality of Scheme 1 is not good and must be improved.

Our Response: A new Scheme is now included.

2. Is it possible to study the progression of the lithiation reaction by dynamic NMR method?

Our Response: In principle yes, it is possible, but we were unable to execute this kind of experiment that was out of the scope of the presented work and would have required special low Temperature NMR equipment.

3. In theoretical calculations, it is unclear why the level of theory was chosen. A reason for choosing this method should be presented here.

Our Response: To corroborate the experimental findings of in-situ FT-IR investigation (Figure 2), a detailed computational investigation using commonly applied methods and basis sets was carried out. An initial screening suggested that there is no substantial difference between several basis sets including 6-311+G(d,p) and 6-31+G(d,p). In addition, there are literature reports which indicate the use of B3LYP/6-31+G(d,p) as the method of choice for the comparative FT-IR study of organic molecules. We thank the reviewer for highlighting this point and to clarify this for the scientific community, a relevant reference related to choosing the level of theory is provided in the supplementary material. Reference: Kerru, N.; Gummidi, L.; Bhaskaruni, S. V. H. S.; Maddila, S. N.; Singh, P.; Jonnalagadda, S. B. Sci. Rep., 2019, 9, 19280.

4. What is the scaling factor for calculated frequency for C=N bond?

Our Response: The calculated vibrational frequencies for C=N bond are unscaled as we observed the value which was very close to the experimental one. In addition, the study was primarily focused on an individual vibrational frequency for C=N bond rather of the entire molecule. This point related to scaling factor is being mentioned at the appropriate place in the manuscript text and for the captions of the Figures 3 and 4.

5. In-situ FT-IR investigation, what is the temperature?

Our Response: The experiment was conducted at -78 °C and this is clearly reported in the manuscript.

6. What is the effect of temperature on epimerization and inversion process? All investigations were done at -78 ˚C.

Our Response: We did not investigate this aspect because above -50 °C the lithiated intermediate decomposes.

7. The authors should be calculated chemical shifts of epimeric hydrogen in all stuctures and compare with experimental data, like computed wavenumber for C=N bond.

Our Response: The computational investigation was just planned to corroborate and provide further insight towards in-situ FT-IR studies while keeping the prime focus of this piece of research on the key aspects related to the development of an efficient -lithiation strategy followed by electrophilic trapping towards enantioenriched azetidines. In addition, we have showcased the synthetic utility of our approach by enantioselective preparation of 2-acylazetdine in an excellent yield and complete preservation of enantioenrichment from starting oxazolinylazetidines to the corresponding products. Calculations of chemical shifts for epimeric hydrogens of lithiated intermediates would not bring additional information because the corresponding experimental 1H chemical shifts are not available (see response for point 2 and refer to Scheme 3 where four different species are expected to be involved).

Reviewer 2 Report

This is an interesting and important paper on azetidine lithiaiton - which provides a nice contrast wiht knonwn work on similar aziridines and related azetidines. The authors have done a thorough and impressive job in exploring this sytem - with React IR, DFT and synthetic/mechanistic results.

I have two main comments:

  1. It is not clear to me whether the authors believe that the 90:10 dr observed with the D quench (and almost other electrophiles) is due to a 90:10 thermodynamic quench of lithiated species or due to a kinetic resolution of a rapidly inverting mistore of diastereomeric lithiated species. This could be made clearer in my view. It was also unclear (although I may have missed it) if the DFT results of the lithiated species supported this thermodynamic mixture. The authors have clearly demonstarated the configurational instability of the lithiated species at -78 °C.
  2.  Related to point 1 above, the result with acetone trapping is very peculiar. One theory I have is that the acetone ratio is reporting on the thermodynamic ratio of diastereomers - as I would expect acetone to be the fastest trapper of all of the electrophiles. Deuterated methanol shoudl of course be fact but it is possible it is slower than a ketone. With the other electrophiles I would expect slow trapping and perhaps some kinetic resolution. It would be good if some of these ideas could be discussed in a revised submission.

Finally, in the electrophile trapping scheme, it would be good to inlcude the electrophile used in each case in the scheme.

Overall, an excellent paper that is suitable for publicaiton after considering the above points.

Author Response

This is an interesting and important paper on azetidine lithiaiton - which provides a nice contrast wiht knonwn work on similar aziridines and related azetidines. The authors have done a thorough and impressive job in exploring this sytem - with React IR, DFT and synthetic/mechanistic results.

I have two main comments:

1. It is not clear to me whether the authors believe that the 90:10 dr observed with the D quench (and almost other electrophiles) is due to a 90:10 thermodynamic quench of lithiated species or due to a kinetic resolution of a rapidly inverting mixture of diastereomeric lithiated species. This could be made clearer in my view. It was also unclear (although I may have missed it) if the DFT results of the lithiated species supported this thermodynamic mixture. The authors have clearly demonstrated the configurational instability of the lithiated species at -78 °C.

Our Response: As reported in Scheme 3, the epimerization process is complicated for the double C-Li and N-R inversion processes. We suppose that four lithiated species, rapidly equilibrating, are involved, thus is difficult to establish a dr in this case.

2. Related to point 1 above, the result with acetone trapping is very peculiar. One theory I have is that the acetone ratio is reporting on the thermodynamic ratio of diastereomers - as I would expect acetone to be the fastest trapper of all of the electrophiles. Deuterated methanol shoudl of course be fact but it is possible it is slower than a ketone. With the other electrophiles I would expect slow trapping and perhaps some kinetic resolution. It would be good if some of these ideas could be discussed in a revised submission.

Our Response: We thank the reviewer for this interesting point. We agree that a kinetic resolution can be operating in this case. We have made this clear in the revised version of the manuscript

Round 2

Reviewer 1 Report

In the revised version, authors considered my comments and suggestions.

It can be accepted in present form.